# Low-Latency Collaborative Predictive Maintenance: Over-the-Air Federated Learning in Noisy Industrial Environments

**DOI:** 10.3390/s23187840

**Published:** 2023-09-12

**Authors:** Ali Bemani, Niclas Björsell

**Affiliations:** Department of Electrical Engineering, Mathematics and Science, University of Gävle, 801 76 Gävle, Sweden; niclas.bjorsell@hig.se

**Keywords:** predictive maintenance, over-the-air federated learning, analog aggregation, low latency, channel noise

## Abstract

The emergence of Industry 4.0 has revolutionized the industrial sector, enabling the development of compact, precise, and interconnected assets. This transformation has not only generated vast amounts of data but also facilitated the migration of learning and optimization processes to edge devices. Consequently, modern industries can effectively leverage this paradigm through distributed learning to define product quality and implement predictive maintenance (PM) strategies. While computing speeds continue to advance rapidly, the latency in communication has emerged as a bottleneck for fast edge learning, particularly in time-sensitive applications such as PM. To address this issue, we explore Federated Learning (FL), a privacy-preserving framework. FL entails updating a global AI model on a parameter server (PS) through aggregation of locally trained models from edge devices. We propose an innovative approach: analog aggregation over-the-air of updates transmitted concurrently over wireless channels. This leverages the waveform-superposition property in multi-access channels, significantly reducing communication latency compared to conventional methods. However, it is vulnerable to performance degradation due to channel properties like noise and fading. In this study, we introduce a method to mitigate the impact of channel noise in FL over-the-air communication and computation (FLOACC). We integrate a novel tracking-based stochastic approximation scheme into a standard federated stochastic variance reduced gradient (FSVRG). This effectively averages out channel noise’s influence, ensuring robust FLOACC performance without increasing transmission power gain. Numerical results confirm our approach’s superior communication efficiency and scalability in various FL scenarios, especially when dealing with noisy channels. Simulation experiments also highlight significant enhancements in prediction accuracy and loss function reduction for analog aggregation in over-the-air FL scenarios.

## 1. Introduction

The pervasive connectivity of numerous devices and sensors, fueled by recent advancements in communication networks and Internet of Things (IoT) applications, has resulted in the generation of immense volumes of data. The copious volume of data at hand serves as the training datasets for machine learning (ML) algorithms that find utility across various applications in industry, such as process optimization, defining product quality, and PM, critical components within the purview of Industry 4.0 [1]. Conventionally, the training process for these ML algorithms has followed a centralized approach, wherein multiple devices transmit their raw and occasionally sensitive data to a PS possessing robust computational resources dedicated to training tasks. Nevertheless, the data required for training these ML algorithms are produced by numerous assets and devices in the industry, which often necessitate privacy-preserving measures to safeguard data integrity. Furthermore, the presence of bandwidth constraints poses a bottleneck when transmitting a massive amount of data from the devices to the PS [2]. For this reason, there has been considerable focus on federated and distributed learning algorithms, primarily due to their ability to train ML models in a fully decentralized manner.

FL algorithms have been proposed as an alternative scheme for privacy-preserving distributed ML, where each device participates in the training process using exclusively locally available data facilitated by a PS [3]. In the context of FL, devices communicate with the PS by exchanging model parameters and their respective local updates, while ensuring that the raw data remain localized on the devices. This approach offers not only privacy advantages but also proves to be a compelling solution for wireless edge devices, particularly when dealing with substantial dataset sizes. In the domain of PM applications, the volume of data obtained from online sensors is significant. Consequently, it becomes imperative to account for this substantial data volume when implementing federated learning techniques for PM. Additionally, time sensitivity and temporal awareness are essential attributes for the effective execution of PM activities. Therefore, time latency should be considered for PM applications.

The migration learning from centralized clouds to the edge enables edge servers to quickly obtain real-time data generated by edge devices, facilitating rapid AI model training. As a result, distributing these models from servers to devices in close proximity enables the devices to respond effectively to real-time events, making them well-suited for PM applications. Despite the rapid advancement of computing speeds, wireless transmission of large amounts of data by any device faces limitations due to limited radio resources and the adverse conditions of wireless channels. This creates a communication bottleneck that hinders fast-edge learning [4,5]. Hence, communication efficiency has been at the forefront of FL, and the paramount objective is to achieve high model accuracy while minimizing the number of communication rounds of resource usage. In the realm of cutting-edge research, the fusion of digital twins and multi-access edge computing has gained significant attention [6]. This innovative approach represents a crucial technology in the context of 6G networks, primarily serving as a fundamental enabler for the Industrial IoT. The primary objective of this research is to minimize the total task completion delay for IoT devices by optimizing various parameters.

Communication schemes for FL can be categorized as digital or analog. Digital communication, though burdensome for wireless networks, assigns communication resources to each client’s ML model parameters. Analog communication reduces overhead by allowing shared resources for transmitting FL models. Early research aimed to reduce communication rounds or payload size. However, most FL literature assumes a perfect communication channel shifting focus to ML design. Recent research addresses this gap by emphasizing system design, especially for wireless FL [7,8,9]. Although early studies have delved deeply into optimizing communication design for FL, the crucial and practical matter of a noisy channel with over-the-air communication and computation (OACC) has not been thoroughly investigated. Incorporating the impact of noisy channels in OACC complicates convergence analysis due to noise propagation during each communication round. Moreover, the collective impact of these noisy communications in OACC on the final learning performance necessitates a comprehensive design and analysis approach.

This work addresses the impact of communication-induced noise during FL training on the convergence and accuracy performance of the ML model, and then proposes robust algorithms within the OACC framework that mitigate these effects and optimize client resources concurrently. Our focus lies in analog communication for model updates [10,11,12] and the exploration of new distributed algorithms capable of withstanding high levels of channel noise and low signal-to-noise ratio (SNR) in industrial environments, in accordance with PM applications.

### 1.1. Related Work

In recent years, substantial endeavors have been undertaken to implement the FL framework over wireless networks, with its origins traced back to McMahan’s seminal works [13,14]. A comprehensive survey of FL has been conducted in [15], offering an overview of this paradigm where statistical models are trained on distributed networks at the edge. These studies explore the distinctive challenges and characteristics of FL compared to traditional approaches while identifying open problems that necessitate interdisciplinary research efforts. However, these works overlook the practical implications of channel effects and assume a seamless integration of FL algorithms into wireless networks. Recently, there has been significant attention given to exploring methods for mitigating the impact of these effects on FL algorithms [16,17,18].

This study primarily focuses on analog aggregation schemes for over-the-air transmission, which are driven by the inherent superposition property of signals in the wireless multiple-access channel. Analog over-the-air aggregation is a highly promising technique extensively used in Federated Averaging (FedAvg) due to the fact that the ps or clients require only the sum of their local gradients or model parameters. These schemes have been explored in various studies, including [2,19], and other relevant works.

In order to provide context and underscore the contributions of this article, we will now discuss previous studies that have investigated FL in the presence of imperfect or noisy communication. The main objective of [20] was to tackle the issue of noise in wireless communications for federated learning. The paper addressed this challenge by formulating the problem using an expectation-based model and a worst-case model. They introduced a sampling-based successive convex approximation algorithm for solving the problem. This approach successfully handles noise by incorporating it as a regularizer in the loss function during the training process. Simulation results showcased improved prediction accuracy and reduced loss function values, affirming the effectiveness of the proposed methods in mitigating the impact of noise. In [21], Amiri et al. examined the impact of a noisy channel on FedAvg. They determined that when dealing with noisy downlink transmission, the presence of noise cannot be neutralized through step-size design, resulting in an inability to ensure precise convergence. Consequently, addressing this fundamental problem has necessitated imposing strict demands on the estimation noise associated with the aggregated model weights.

In [22], researchers conducted an investigation into the influence of a noisy channel in uplink and downlink analog transmission on the training process. The findings of this analysis led the authors to conclude that for the FedAvg algorithm to converge, it is necessary for the noise variance to decrease at a rate of $O\left(1/k^{2}\right)$, where *k* represents the communication round. Consequently, in order to effectively mitigate the impact of channel noise on the training procedure, the authors recommend two approaches. Firstly, it is advised to employ an increased transmission power gain of $O\left(k\right)$ in both uplink and downlink transmissions. Alternatively, if the power gain remains fixed, extending the transmission time to O(k) is suggested. The implementation of either of these strategies is crucial for preserving the integrity of the training process when faced with channel noise.

Another critical issue is analog transmission, which is widely used in over-the-air aggregation within the wireless channel. It is extensively utilized and plays a crucial role in enhancing spectral efficiency and reducing multi-access latency. Gau et al. [23] developed a hardware transceiver and application software to train a real-world FL task using over-the-air analog aggregation. They focused on developing an over-the-air aggregation solution for wireless FL based on orthogonal frequency-division multiplexing (OFDM). The main challenge they faced was achieving perfect waveform superposition, which was complicated by the presence of frame timing offset and carrier frequency offset. To tackle these challenges, they proposed a two-stage waveform pre-equalization technique.

The focus of this study pertains to the exploration of algorithms aimed at resolving optimization problems through over-the-air analog aggregation. While numerous papers have discussed communication-efficient solutions for distributed learning problems, it is of paramount importance to thoroughly investigate the optimization of analog FL problems over noisy communication channels. This particular problem presents significant importance and complexity, necessitating meticulous attention and consideration.

### 1.2. Our Contributions

The major contributions of this work are summarized as follows.
We propose a hierarchical approach to PM, building upon our prior work [1]. The key distinction lies in the utilization of OACC for the FL algorithm at the factory level. This choice is motivated by the benefits of low latency, making it suitable for PM applications while also improving spectral efficiency. At higher levels, such as fog and cloud servers, occasional requests are made to the factory level for the aggregate model, enabling averaging over multiple factory parameters, FedAvg. Our primary emphasis is on the factory level, specifically investigating FLOACC as the focal point of our research.We propose FSVRG-OACC as a distributed approach to solve the optimization problem for PM at the factory level based on OACC. FSVRG-OACC leverages analog over-the-air aggregation, which enables it to effectively handle highly noisy communication channels and allows for improved convergence in minimizing the cost function associated with the ML algorithm.FSVRG-OACC facilitates the transmission of local gradient updates by individual agents, capitalizing on the advantages of computation over the air. This algorithm effectively mitigates the impact of channel perturbations on convergence by incorporating the effects of the communication channel into the algorithm update process. The utilization of FSVRG-OACC ensures that convergence is not compromised, enabling efficient and robust optimization in the presence of varying channel conditions.The simulation results demonstrate the substantial reduction in convergence sensitivity to noise achieved by our proposed algorithm. This finding holds significant implications for the implementation of ML algorithms on analog over-the-air aggregation in highly noisy industrial environments.

The remainder of this paper is structured as follows. In Section 2, we introduce the system model, starting with the description of FL over-the-air analog aggregation and extending it to FLOACC. Section 3 presents our proposed algorithm, FSVRG-OACC, along with a comparison to other stochastic gradient descent algorithms. Following that, in Section 4, we evaluate the performance of FSVRG-OACC through a PM application and present the corresponding experimental results. Finally, in Section 5, we provide concluding remarks and discuss future research directions.

## 2. System Model

### 2.1. Federated Edge Learning System

We consider a distributed learning system specifically designed for a PM application at the factory level. This system comprises a single parameter server and K edge nodes, as depicted in Figure 1. Collaboratively, the edge nodes at each factory train a shared learning process involving the global model **w**. Each machine collects a fraction of labeled training data via the interaction with a local dataset, denoted as $D_{1},D_{2},\ldots ,D_{K}$. The local loss function of the model vector **w** on $D_{k}$ is given by
(1)$$\begin{array}{c} F_{k}\left(w\right)=\frac{1}{\phi_{k}}\sum_{\left(x_{j},y_{j}\right)\in D_{k}}f\left(w,x_{j},y_{j}\right), \end{array}$$
where $\phi_{k}$ is the number of data points stored on data partition $D_{k}$ and $f(w,x_{j},y_{j})$ is the loss function quantifying the prediction error of the model **w** for each data sample *j*, which consists of the training sample $x_{j}$ and its ground true label $y_{j}$. For convenience, we write $f(w,x_{j},y_{j})$ as $f_{j}\left(w\right)$ and assume uniform sizes for local datasets $\phi_{k}$ for all *k*. Table 1 presents the typical loss function used in FL for various applications. In the specific case of anomaly detection over edge nodes at the factory, the squared-SVM loss function has been employed in this study as a leading ML method that offers flexibility in modeling complex nonlinear boundaries between normal and abnormal data points. With these definitions, the global loss function on all the distributed datasets can be defined as
(2)$$\begin{array}{c} F\left(w\right)=\frac{\sum_{j\in \cup_{k}D_{k}}f_{j}\left(w\right)}{\cup_{k}\phi_{k}}=\frac{1}{K}\sum_{k=1}^{K}F_{k}\left(w\right), \end{array}$$
where $\left|.\right|$ denotes the size of the datasets and each dataset satisfies $D_{i}\cap D_{j}=\emptyset$ when i≠j. The training target is to minimize the global loss function F(w) according to the distributed process to find $w^{*}$.
(3)$$\begin{array}{c} w^{*}=\arg\min F\left(w\right) \end{array}$$

In this section, we begin by presenting the problem formulation of FL in the context of a hierarchical PM scenario. Subsequently, we delve into the specific case of FL-OACC, where all agents at the factory levels broadcast their updated models over the air, leveraging the advantages of computation through analog communication.

To compute F(w), one method involves directly uploading all local data, which raises privacy concerns. To address this issue, the FL framework is employed to solve the problem outlined in Equation (Equation 3) through a distributed approach. There are two approaches based on FedAvg for solving this distributed optimization problem:Model averaging: In this approach, each agent minimizes its local loss function and transmits the model parameters to the PS for aggregation. In the second round of iteration, the agent receives the updated model from the PS.Gradient averaging: In this approach, each agent calculates the gradient of its loss function and transmits the gradient to the PS for aggregation. In the second round of iteration, the agent can update its model based on the gradient averaging received from the PS.

The agents generally employ a gradient descent (GD) algorithm or stochastic gradient descent (SGD) algorithm to minimize the loss function described in Equation (Equation 1). A single-step SGD for updating the model parameter for device *K* can be defined following the model and gradient averaging, respectively.
(4)$$\begin{array}{c} w_{k}\left[n+1\right]=w\left[n\right]-\alpha \nabla F_{k}\left(w\left[n\right]\right), \end{array}$$
(5)$$\begin{array}{c} w_{k}\left[n+1\right]=w_{k}\left[n\right]-\alpha \nabla F\left(w\left[n\right]\right), \end{array}$$
where α is the step size and ∇ is the gradient operator. Hence, the only operation performed by the PS is to compute the average of the model parameters or the gradients received from the agents.
(6)$$\begin{array}{c} w\left[n+1\right]=\frac{1}{K}\sum_{k=1}^{K}w_{k}\left[n+1\right], \end{array}$$
(7)$$\begin{array}{c} \nabla F\left(w\left[n\right]\right)=\frac{1}{K}\sum_{k=1}^{K}\nabla F_{k}\left(w_{k}\left[n\right]\right). \end{array}$$

The learning process entails iterating between Equations (Equation 4) and (Equation 6) or Equations (Equation 5) and (Equation 7) until the model converges. The averaging process on the PS serves as a motivation for the low-latency FL scheme, utilizing FLOACC. Further details about FLOACC are provided below.

### 2.2. Fl Over-the-Air Communication and Computation

In this section, we discuss the details of Federated Learning over-the-Air Communication and Computation, abbreviated as FLOACC. This method provides an efficient multi-access scheme in a low-latency scenario, which is crucial for applications like PM that require very fast and real-time task decision making. The idea of the over-the-air computation model for FL has been examined in several previous works, such as [8,16]. Their approach was inspired by the PS’s lack of interest in individual model weight vectors. Instead, the server solely needs the average of the model weights, which is automatically provided by the wireless multiple-access channel in the form of their sum.

As depicted in Figure 2, the FLOACC enables simultaneous transmission of result vectors from all devices and assets at the factory level to the PS in an analog manner. Let $w_{k}={\left[w_{k,1},\cdots ,w_{k,q}\right]}^{T}$ denote q×1 local model parameter vector and $\nabla F_{k}\left(w\right)={\left[\nabla F_{k,1},\cdots ,\nabla F_{k,q}\right]}^{T}$ be the local gradient vector of the loss function from the *k*-th device. In FLOACC, it is assumed that the local gradient of each model is transferred over an analog medium. In this scenario, the transmitted symbols are denoted by $\left\{{\tilde{\nabla F}}_{k,i}\right\}$ and are normalized to have zero mean and unit variance $E\left({\tilde{\nabla F}}_{k,i}{\tilde{\nabla F}}_{k,i}^{*}\right)=1$. By employing OFDM, it becomes feasible to allocate each element of the gradient vector to a distinct sub-carrier OFDM channel. This approach enables a significant reduction in the learning process latency for FLOACC.

During each round *n*, all the local devices at a factory simultaneously transmit their local gradient based on the distributed loss function as $h_{k}\left[n\right]p_{k}\left[n\right]\nabla F_{k}\left(w\left[n\right]\right)$, where $h_{k}\left[n\right]∼CN\left(0,1\right)$ is the small-scale fading of the channel between the *k*-th device and the PS, and $p_{k}\left[n\right]$ is the allocated transmission power for each device *k*. In particular, the aggregated gradient in the *n*-th communication round, denoted by y[n], is expressed as follows.
(8)$$\begin{array}{c} \hat{\nabla F}\left(w\left[n\right]\right)=\sum_{k=1}^{K}h_{k}\left[n\right]p_{k}\left[n\right]\nabla F_{k}\left(w_{k}\left[n\right]\right)+Z\left[n\right], \end{array}$$
where $Z\left[n\right]∼CN\left(0,\sigma_{z}^{2}I\right)$ is additive white Gaussian noise. Note that in over-the-air analog aggregation, it is possible to transfer the model parameters as well. However, in FLOACC, we only consider the local gradient during aggregation. This is because the aggregated gradient is less sensitive in the optimization algorithms compared to the model parameters. We will discuss this issue further in the next section.

We assumed that, similar to the LTE system, there are common downlink reference symbols in the radio resource block for the devices at the factory level to estimate the channel fading coefficient. Then, the local devices can transmit the signal with the specific power they calculated. The received signal at the PS has been depicted with gain blocks in Figure 3, so the PS can estimate the aggregated gradient as follows:(9)$$\begin{array}{c} y\left[n\right]=\sum_{k=1}^{K}\frac{h_{k}\left[n\right]p_{k}\left[n\right]\nabla F_{k}\left(w_{k}\left[n\right]\right)}{\sqrt{\eta }}+\frac{Z[n]}{\sqrt{\eta }}, \end{array}$$
where η is a receiver scaling factor. Owing to the imposed physical constraints, the transmission of each device is subject to a long-term transmission power constraint.
(10)$$\begin{array}{c} E\;\left|\left[\sum_{n=1}^{N}{\left|p\left[n\right]\left(h[n]\right)\right|}^{2}\right]\right|\leq P_{0}. \end{array}$$

### 2.3. Effective Noise and Definition of SNR in FLOACC

In order to keep the problem general, we adopt the effective noise model in analog aggregation. We assume accurate channel state information at the transmitters. To meet the aggregation requirement of FL, the local devices implement the channel inversion rule, which yields the instantaneous transmit power of user *k* during the communication round *n* for gradient aggregation.
(11)$$\begin{array}{c} p_{k}\left[n\right]=\frac{h_{k}{\left[n\right]}^{H}}{\left|h_{k}\left[n\right]\right|}\cdot \frac{\sqrt{\eta }}{\left|h_{k}\left[n\right]\right|}, \end{array}$$
where *H* is Hermitian transpose. Based on this definition, the received signal can be formulated as follows.
(12)$$\begin{array}{c} y\left[n\right]=\sum_{k=1}^{K}\nabla F_{k}\left(w_{k}\left[n\right]\right)+\frac{Z[n]}{\sqrt{\eta }}. \end{array}$$

η represents a scalar that signifies the average transmission power, based on which the received SNR of the global gradient can be expressed as follows.
(13)$$\begin{array}{c} \mathrm{SNR}\left[n\right]=E{∥\frac{1}{K}\sum_{i=1}^{q}\frac{\sqrt{\eta }\sum_{k\in K}\nabla F_{k}\left(w_{k,i}\left[n\right]\right)}{Z_{i}\left[n\right]}∥}^{2}=\frac{\eta E{∥\sum_{k\in K}\nabla F_{k}\left(w_{k}\left[n\right]\right)∥}^{2}}{\sigma_{z}^{2}K^{2}}. \end{array}$$

It is evident that the received SNR is consistent across all users due to devices with weaker channels compensating by transmitting at higher powers. In various studies, devices with significantly weak channels are often excluded from training due to their inability to pre-equalize their channels. Many research has been undertaken to optimize η to enhance model convergence over the wireless communication network. In [24,25], the optimal selection of $p_{k}$ and η is determined by solving the corresponding optimization problem.
(14)$$\begin{array}{c} \mathrm{MSE}=E{∥y\left[n\right]-\sum_{k=1}^{K}\nabla F_{k}\left(w_{k}\left[n\right]\right)∥}^{2}=\frac{1}{K^{2}}\sum_{k=1}^{K}{\left(\frac{\left|h_{k}\right|p_{k}}{\sqrt{\eta }}-1\right)}^{2}+\frac{\sigma_{z}^{2}}{\eta K^{2}}, \end{array}$$
(15)$$\begin{array}{cc} \min_{p_{1},p_{2},\ldots ,p_{K},\eta } & \frac{1}{K^{2}}\sum_{k=1}^{K}{\left(\frac{\left|h_{k}\right|\sqrt{p_{k}}}{\sqrt{\eta }}-1\right)}^{2}+\frac{\sigma_{z}^{2}}{\eta K^{2}}\\ \;s.t.\; & p_{k}\leq P_{\max},\forall k\in \left\{1,2,\ldots ,K\right\}. \end{array}$$

Extensive research has been conducted to optimize transmit power for the purpose of achieving model convergence and mitigating the effects of existing noise in the wireless transmission medium. However, to the best of our knowledge, no existing studies have investigated the comparison of convergence performance among different algorithms without any transmission power control. In the upcoming section, we delve into various gradient descent algorithms for ML optimization problems, considering the over-the-air scheme, and provide an analysis of which algorithm demonstrates superior convergence properties.

## 3. Proposed Adaptive FSVRG-OACC Algorithm

In this section, our specific focus lies on algorithms suitable for solving problem (Equation 3) within the context of over-the-air and analog aggregation. Firstly, in Section 3.1, we examine baseline algorithms that are compatible with distributed problems and analog gradient transmission (GT), while also highlighting the distinctions between model transmission (MT) and GT. Next, in Section 3.2, we delve into randomized methods that, in an initial approximation, integrate the advantages of cost-effective iterations from SGD with the rapid convergence of GD. Many of these methods can be categorized into one of two classes: dual methods of the randomized coordinate descent type and primal methods of the stochastic gradient descent with variance reduction type. Our emphasis lies on stochastic variance reduced gradient (SVRG), and we optimize this method for FL within the framework of analog aggregation while considering the presence of white Gaussian channel noise.

### 3.1. Baseline Algorithms

A fundamental approach for solving a distributed optimization problem with the structure (Equation 4) involves employing the GD algorithm, particularly when the functions *f* possess smoothness and convexity characteristics. In a distributed setting, there are two possible approaches: the GT method, where gradients are transmitted over the air and an aggregated gradient signal is received, which is subject to noise for model updating using GD, or the MT method, where the model is updated through GD iterations and then transmitted for model aggregation. In the presence of noise, the convergence deteriorates due to the elevated sensitivity of the loss function to the model parameters. Therefore, in over-the-air distributed algorithms, the GT approach is preferred. This preference stems from the lower sensitivity of the cost function to the aggregated gradient compared to the individual model parameters. GD demonstrates a fast convergence rate. However, each iteration has the potential to be computationally intensive on each local device. In contrast, SGD selects a random data label and performs the update, which offers a more efficient alternative.

### 3.2. Fsvrg-Oacc Algorithm

An additional algorithm within the SGD category is SVRG [26]. The SVRG algorithm operates through two nested loops. The outer loop involves calculating the full gradient of the entire function, $\nabla F(w^{t}\left[n\right])$, which is typically a computationally expensive operation to be avoided whenever possible. In the inner loop, the update step is iteratively computed as follows.
(16)$$\begin{array}{c} w\left[n+1\right]=w\left[n\right]-\alpha \left[\nabla F^{i}\left(w\left[n\right]\right)-\nabla F^{i}\left(w^{t}\left[n\right]\right)+\nabla F\left(w^{t}\left[n\right]\right)\right], \end{array}$$
where $\nabla F^{i}\left(w\left[n\right]\right)$ represents the stochastic gradient computed based on a randomly selected data label, $\nabla F^{i}\left(w^{t}\left[n\right]\right)$ denotes the stochastic gradient computed over the entire dataset, and α is stepsize. This iteration is specific to a single device, and its fundamental concept lies in utilizing stochastic gradients to estimate the gradient change from point $w^{t}$ to w, rather than directly estimating the gradient itself.

Indeed, this algorithm is naturally suited for centralized implementations since it necessitates computing the stochastic gradient over the complete dataset, thereby making it well-suited for centralized scenarios. But a notable contribution was made in [27], where they introduced FSVRG, which is particularly applicable in the context of distributed optimization. They demonstrated that existing SVRG algorithms are not suitable for distributed approaches and proposed the FSVRG algorithm, specifically designed for sparse distributed convex problems. The pseudocode for FSVRG is provided in Algorithm 1. This algorithm has been implemented and subjected to evaluation, and the results will be presented in the experimental section. The findings indicate that this algorithm does not demonstrate satisfactory convergence in the realm of FL for over-the-air analog aggregation, specifically in relation to the absence of power transfer control.

Let us now elucidate the motivation behind considering a different algorithm suitable for FL in the context of over-the-air analog aggregation. A crucial aspect that demands attention is the significant variation in the number of available data points among different devices, which may differ greatly from the average number of data points available to any single device. It looks like a similar issue to FSVR, but it should be noted that in our assumptions, analog communication is the sole communication type between local devices and the PS. As a result, the PS lacks information regarding the number of data points and the type of data distribution.
**Algorithm 1:** Federated SVRG
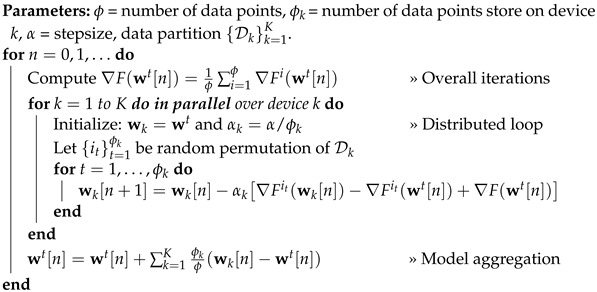


Additionally, this scenario frequently entails the local data being clustered around a specific pattern, which renders it unrepresentative of the overall distribution we aim to learn. Consequently, considering an aggregation on the entire gradient direction in each iteration could be a promising approach that could be undertaken in the concept of analog aggregation.

From a practical perspective, in FSVRG-OACC, it is postulated that all devices possess a randomly allocated initialization value for the parameter vector, $w_{k}$. This assumption holds significant importance in the practical execution of the algorithm. The proposed algorithm involves two communication rounds, which results in increased communication costs. However, it offers advantages in terms of the convergence algorithm. Algorithm 2 introduces the FSVRG-OACC, a modified FSVRG variant tailored for over-the-air analog aggregation.

During the initial communication round (distributed loop 1), all devices compute the complete gradient of the entire function and subsequently determine the internal gradient as $g_{k}=\left[\nabla F_{k}^{i_{t}}\left(w^{t}\left[n\right]\right)-\nabla F_{k}\left(w\left[n\right]\right)\right]$, where $i_{t}$ is sampled uniformly from the local dataset $D_{k}$. These gradients are derived using SGD, rendering their computation relatively inexpensive. The computed internal gradient is then transmitted over the air to the PS, while the estimated aggregated internal gradient is sent back to the devices via the analog medium, as shown in Figure 2.
(17)$$\begin{array}{c} \hat{g}=\frac{1}{K}\sum_{k=1}^{K}\frac{h_{k}\left[n\right]p_{k}\left[n\right]g_{k}}{\sqrt{\eta }}+\frac{z[n]}{\sqrt{\eta }}. \end{array}$$

The estimated aggregated internal gradient, denoted as $\hat{g}$, compels all devices in the second round of communication (Distributed loop 2) to move in the same direction. In this communication round, the updated gradient is denoted as $\hat{G}$, which was estimated in the first round as $\hat{G}=\nabla F^{i_{t}}\left(w_{k}\left[n\right]\right)-\hat{g}$ for device k. Subsequently, the PS aggregates this gradient over the air from all devices and transmits it back to them for the remaining iterations. Each device *k* uploads the following gradient over the air for aggregation.
(18)$$\begin{array}{c} G_{k}=\nabla F^{i_{t}}\left(w_{k}\left[n\right]\right)-\hat{g}, \end{array}$$
(19)$$\begin{array}{c} G_{k}=\nabla F^{i_{t}}\left(w_{k}\left[n\right]\right)-\left[\frac{1}{K}\sum_{k=1}^{K}\frac{h_{k}\left[n\right]p_{k}\left[n\right]\left[\nabla F_{k}^{i_{t}}\left(w^{t}\left[n\right]\right)-\nabla F_{k}\left(w\left[n\right]\right)\right]}{\sqrt{\eta }}+\frac{z[n]}{\sqrt{\eta }}\right]. \end{array}$$

**Algorithm 2:** FSVRG With Over-the-Air Communication and Computation (FSVRG-OACC)

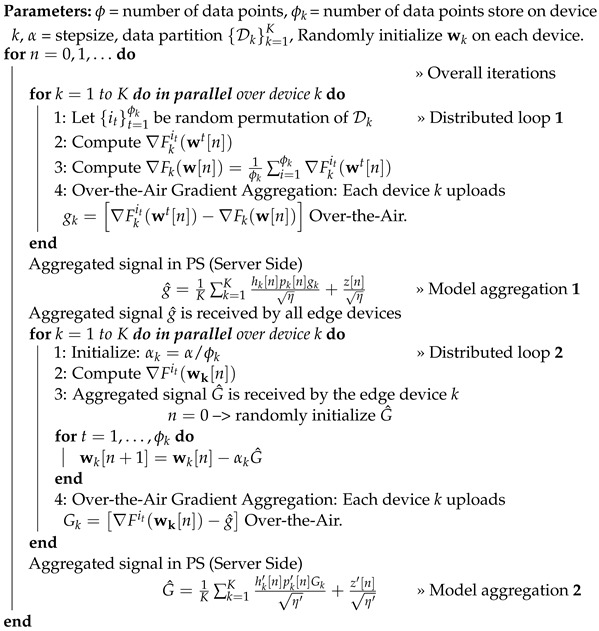



By aggregating the gradient $G_{k}$ at the PS and transmitting it back to each device, a uniform descent direction can be achieved across all devices.
(20)$$\begin{array}{c} \hat{G}=\frac{1}{K}\sum_{k=1}^{K}\frac{h_{k}^{\prime }\left[n\right]p_{k}^{\prime }\left[n\right]G_{k}}{\sqrt{\eta^{\prime }}}+\frac{z^{\prime }\left[n\right]}{\sqrt{\eta^{\prime }}}, \end{array}$$
(21)$$\begin{array}{c} \hat{G}=\frac{1}{K}\sum_{k=1}^{K}\frac{h_{k}^{\prime }\left[n\right]p_{k}^{\prime }\left[n\right]\left[\nabla F^{i_{t}}\left(w_{k}\left[n\right]\right)-\left[\frac{1}{K}\sum_{k=1}^{K}\frac{h_{k}\left[n\right]p_{k}\left[n\right]\left[\nabla F_{k}^{i_{t}}\left(w^{t}\left[n\right]\right)-\nabla F_{k}\left(w\left[n\right]\right)\right]}{\sqrt{\eta }}+\frac{z[n]}{\sqrt{\eta }}\right]\right]}{\sqrt{\eta^{\prime }}}+\frac{z^{\prime }\left[n\right]}{\sqrt{\eta^{\prime }}}. \end{array}$$

This approach is motivated by the primary goal of maintaining gradient step consistency among all clients by utilizing the aggregated gradient over-the-air signal in stochastic first-order methods. Therefore, the algorithm’s complexity is exceptionally low due to the simplicity of first-order gradient calculations at each step. The only cost involved is the number of communication rounds.

In summary, the gradient update $\left[\nabla F^{i_{t}}\left(w_{k}\left[n\right]\right)-\nabla F_{k}^{i_{t}}\left(w^{t}\left[n\right]\right)+\nabla F_{k}\left(w^{t}\left[n\right]\right)\right]$ is divided into two parts for FSVRG-OACC. In the first distributed loop, the last two gradients $\nabla F_{k}^{i_{t}}\left(w^{t}\left[n\right]\right)$ and $\nabla F_{k}\left(w^{t}\left[n\right]\right)$ are calculated, and afterward, the distance between these two gradients is aggregated across all edge devices. In the second distributed loop, we can access the aggregated value of $\left[\nabla F_{k}^{i_{t}}\left(w^{t}\left[n\right]\right)-\nabla F_{k}\left(w^{t}\left[n\right]\right)\right]$ for all devices. In the first iteration of this loop (n=1), we calculate the stochastic gradient for each edge device and subtract it from the aggregated gradient obtained from the first distributed loop $\left[\nabla F^{i_{t}}\left(w_{k}\left[n\right]\right)-\mathrm{aggregation}\left(\nabla F_{k}^{i_{t}}\left(w^{t}\left[n\right]\right)-\nabla F_{k}\left(w^{t}\left[n\right]\right)\right)\right]$. This gives us $G_{k}$, the overall gradient update for device k, which is then aggregated with all other gradients over the air. Starting from the second iteration (n=2) and beyond, we can utilize the whole aggregated gradient $\hat{G}$ to directly update the model parameters of each device.

## 4. Performance Evaluation of FSVRG-OACC

In this section, we evaluate the performance of the proposed algorithm on the task of anomaly detection for a PM application in an FL manner with over-the-air analog aggregation. It aims to investigate the convergence characteristics of four optimization algorithms, GD, SGD, FSVRG, and FSVRG-OACC, in the context of over-the-air analog aggregation. The primary focus is to analyze the performance of these algorithms in terms of both model accuracy and convergence time, which are crucial performance metrics in the field. Similar to prior research in the domain of Federated Machine Learning Algorithm for Collaborative Predictive Maintenance [1], we employ the widely recognized and benchmarked CMAPSS [28] dataset in this study.

The C-MAPSS model, developed by NASA and implemented in MATLAB/Simulink, represents a nonlinear dynamic model of a commercial turbofan engine. By manipulating the input parameters of this simulation model, it becomes possible to simulate diverse degradation patterns under various engine conditions. In order to generate datasets reflecting different fault modes, four distinct time series (FD001, FD002, FD003, FD004) were produced using these simulation tools. These datasets comprise multivariate time series, each of which is further divided into training and testing subsets. For the purpose of this study, we selected and analyzed two datasets: FD003 and FD004. The FD003 dataset comprises 100 test trajectories and 100 train trajectories, focusing on a single fault mode and two types of degradation: high-pressure compressor degradation and fan degradation. On the other hand, the FD004 dataset includes 248 test trajectories and 249 train trajectories, covering six fault modes and two types of degradation. The time series data in each dataset include 21 sensor observations, three operating settings, a trajectory ID, and a cycle count. The Remaining Useful Life (RUL) of an engine is estimated based on the number of operation cycles remaining before the engine fails.

### 4.1. Algorithm Implementation

In this study, a federated SVM model was employed to predict the RUL using the provided time series data. Specifically, FD003 and FD004 datasets were distributed among ten devices located at the factory site, with the purpose of participating in a collaborative PM scenario. It is crucial to emphasize that the data distribution was non-independent and not identically distributed (non-IID). This implies that the instances of failures were not randomly distributed among the edge devices, and the data distribution was not evenly spread across all edge devices. Instead, specific edge devices were linked to particular types of failures, while others experienced different types. We intentionally opted for this non-IID distribution as a worst-case scenario to thoroughly assess the effectiveness of our proposed method. This collaboration aimed to achieve low latency through over-the-air analog aggregation. During all the simulations, the optimizer parameters were configured to achieve optimal performance. In particular, the learning is kept fixed at 0.01, and the $\ell_{2}$-regularization parameter λ is set to λ=0.1. The momentum factor is kept equal to 0.9 and the number of local epochs is set to 1. The definition of the loss function for the federated SVM is as follows.
(22)$$\begin{array}{c} f\left(w\right)=\frac{1}{\phi_{k}}\sum_{j\in D_{k}}f_{j}\left(w\right)+\lambda {∥w∥}_{2}^{2}\\ f_{j}\left(w\right)=\max\left(0,1-y_{j}w^{T}x_{j}\right), \end{array}$$

The input data, denoted as $x_{j}$, are structured in a 2D format resembling an image, where one dimension corresponds to the sequence length, and the other dimension represents the number of sensor measurements. The output variable, $y_{j}$, indicates the condition of the engine. Specifically, if the input data are associated with a particular condition whose RUL is below a specified threshold, the output value is assigned as {−1}, indicating the detection of an anomaly. Conversely, if the RUL exceeds the threshold, the output value is assigned as {1}. In all the conducted experiments, the reported convergence and accuracy averaged over five independent runs were calculated over 500 communication rounds.

### 4.2. Performance of FSVRG-OACC

#### 4.2.1. General Performance

We first conduct a comparison between FSVRG-OACC and standard FSRVG, SGD, and GD methods to illustrate the notable enhancements in convergence rate and accuracy achieved in the context of over-the-air analog aggregation with a noisy environment. Throughout the experiments, we assessed the performance of each method and observed the significant gains achieved by FSVRG-OACC. We assume the environment noise variance to be $\sigma_{z}^{2}=1$ for $Z_{k}∼\mathrm{CN}\left(0,\sigma_{z}^{2}I\right)$, and the probability of noise presence as p=1. In this study, our primary focus is not on the transmission policy. Therefore, we make the assumption that the channel inversion policy is employed for estimation. Additionally, we assume that each component of Z[n] has a zero mean and a variance of $\sigma_{z}^{2}$.

The performance is demonstrated on the FD003 and FD004 datasets. In this experiment, for all of K=10 devices at the factory site, this implies that the transmitting power is set to *P* = 300 mW and the receiving scale factor, $\sqrt{\eta }$, is set to 0.1. The accuracy formula used is as follows:(23)$$\begin{array}{c} \;\mathrm{accuracy}\;=\frac{\mu_{P}+\mu_{N}}{\mu_{P}+\mu_{N}+\Gamma_{P}+\Gamma_{N}}, \end{array}$$
where $\mu_{P}$ represents the number of true positives, $\Gamma_{P}$ represents false positives, $\mu_{N}$ represents true negatives, and $\Gamma_{N}$ represents false negatives. The results have been plotted in Figure 4 for FD003 and Figure 5 for FD004. We observed the following results.
In the case of FD003, it is evident that both the GD-MT and FSVRG algorithms fail to converge under the given noise environment and transmission power settings. Consequently, these algorithms are excluded from the accuracy analysis. On the other hand, the GD-GT and SGD-GT algorithms demonstrate convergence, but they exhibit significant fluctuations during the convergence phase. In contrast, our proposed FSVRG-OACC method shows excellent convergence performance under the same environmental conditions and noise levels.As depicted in the accuracy plot for FD003, both SGD-DT and GD-GT algorithms experience a considerable drop in accuracy. However, our proposed algorithm achieves an average accuracy of 91% and demonstrates higher stability compared to the other algorithms.In the case of FD004, we observed similar results, although this dataset presents greater challenges due to its inclusion of six fault modes, making the prediction algorithm significantly more complex compared to other CMAPSS datasets. Despite these difficulties, our proposed FSVRG-OACC algorithm demonstrates robust convergence, with only minimal fluctuations occurring after the convergence stage. These fluctuations can be attributed to the estimation anomalies present in the most challenging instances of the dataset.In the accuracy plot, our proposed method achieved a commendable accuracy of 61% on the model. In contrast, the average accuracy of the other two converged methods is lower than that of the proposed method, and their accuracy values exhibited less fluctuation during the communication rounds.

In conclusion, we provide the average local training runtime and accuracy of FSVRG-OACC, SGD-GT, and GD-GT at the client level, as summarized in Table 2. It is noteworthy to mention that the computational runtime is primarily governed by the computation of gradients in all 500 iterations. Additionally, we assume that the communication round is negligible in comparison to the gradient computation. As is evident, the runtime in FSVRG-OACC exceeds that of the other algorithms. However, this method, employed in FLOACC, demonstrates substantial reduction in communication latency while achieving commendable convergence and consistent accuracy.

We compared our results with a scenario where there is no communication noise. In a previous study, we looked at how well the FL algorithm works for PM applications when there is no communication interference. We summarized those results in Table 3. As evident from the results, there is a significant disparity in accuracy between GD-GT and SGD-GT when applied to FLOACC and noiseless communication settings. This demonstrates that employing a power transmission policy can enhance the accuracy of both GD and SGD in the context of FLOACC, bringing them closer to achieving results similar to those in noiseless communication scenarios. It is worth noting that, for the FSVRG-OACC method and FSVRG applied to noiseless communication, the difference in accuracy remains minimal. This observation is particularly noteworthy in the case of FD003, where the RUL prediction task is less challenging than FD004. These findings underscore the robustness and effectiveness of our proposed approach, which achieves these results without the need for any power transmission policies.

#### 4.2.2. Performance of FSVRG-OACC with Varying Noise Level

In this section, we conduct an analysis to assess the robustness of the proposed algorithms for Over-The-Air analog aggregation in the presence of varying levels of noise. The performance evaluation is conducted using a part of the FD003 dataset while systematically varying the noise level $\sigma_{z}$. Specifically, we consider noise levels $\sigma_{z}$ from the set {0.25,0.75,1.5,2.5} to assess the algorithm’s performance under different noise conditions. For the total of 10 local devices, the receiver scaling factor, denoted as $\sqrt{\eta }$, is varied across different values {0.4,0.13,0.06,0.04}. To investigate the impact of noise, we manipulate the probability of noise presence, denoted as *p*, throughout the experiments. The set {0.1,0.25,0.5,1} is utilized to vary the probability of noise. The results for GD-GT and SGD-GT have been plotted in Figure 6 and Figure 7.

For both GD-GT and SGD-GT, as the noise level increases, there is a significant degradation in performance, accompanied by amplified fluctuations in the loss function. Additionally, we observe a parallel trend in the cost function when the probability of noise presence is elevated. Furthermore, we conducted the same analysis on the FSVRG-OACC method, and the corresponding results are presented in Figure 8. Notably, the proposed algorithm demonstrates significantly enhanced robustness against higher noise levels and increased probability of noise presence.

All of these analyses provide evidence supporting the suitability of the proposed FSVRG-OACC method for analog aggregation and FL over the air. It is important to note that all of these analyses were conducted under the same transmission power conditions. However, we anticipate that employing a power control policy in communication with this method would yield even more accurate results in convergence and model accuracy.

On the other hand, implementing our proposed approach in real-world industrial settings has practical implications and challenges. A significant part of this approach is adjusting the local gradients in large edge devices using an analog waveform and sending them through the same wireless channels. Our main challenge is achieving perfect waveform superposition, which is crucial for our algorithm’s success. This task is complicated because of frame timing offset and carrier frequency offset. To overcome these issues, we require high-performance devices with substantial computational capabilities. Another important challenge emerges when we only have access to partial information, resulting in some edge devices being unable to effectively join the FLOACC aggregation process. This led to a deviation in the gradient descent regime. This deviation highlights the necessity for robust device selection and strategies to manage situations where certain devices may have limited participation capabilities.

## 5. Conclusions

In this paper, our focus was on FL scenarios that leverage wireless transmission channels for both communication and computation. The design presented capitalizes on the waveform superposition property inherent in a multi-access channel, which enables efficient update aggregation, optimizing the communication process. We specifically emphasized the potential of over-the-air analog aggregation in FL for hierarchical PM scenarios. Our study focused on the factory site as the lower hierarchical component, recognizing the criticality of meeting low-latency requirements for time-sensitive applications. We have demonstrated that this challenge can be effectively addressed through the utilization of analog aggregation.

Throughout our investigation, we thoroughly examined the impact of practical channel effects, including noise and fading, on the learning algorithm’s performance. To enhance the robustness of learning algorithms against channel noise effects, we proposed a novel algorithm called FSVRG-OACC. The effectiveness and superiority of the proposed algorithm were demonstrated through the application of a distributed SVM for anomaly detection using the CMPASS dataset. The simulation results validate the effectiveness of FSVRG-OACC in reducing aggregation distortion while operating at the same transmission power level. Furthermore, the learning behavior of the proposed algorithm can be further enhanced by incorporating power control and device selection policies into its design.

## Figures and Tables

**Figure 1 sensors-23-07840-f001:**
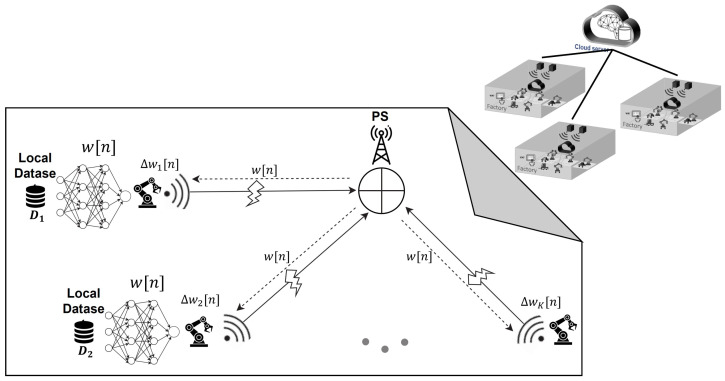
FL edge system model for a hierarchical PM scenario at the factory level.

**Figure 2 sensors-23-07840-f002:**
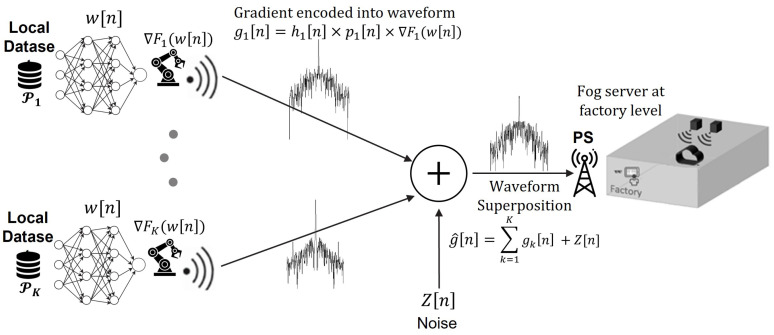
FLOACC scenario at the factory level.

**Figure 3 sensors-23-07840-f003:**
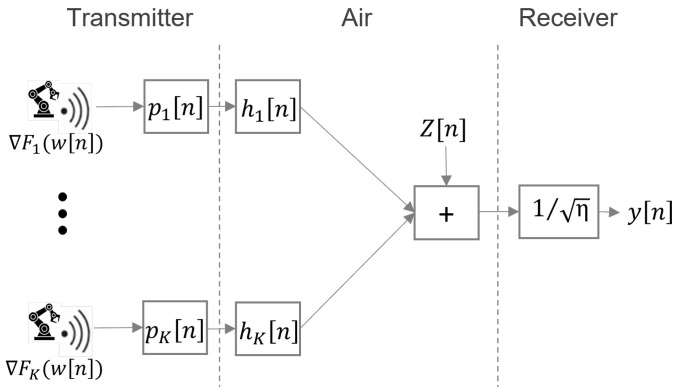
Illustration of the system design for FLOACC using analog transmission over-the-air aggregation.

**Figure 4 sensors-23-07840-f004:**
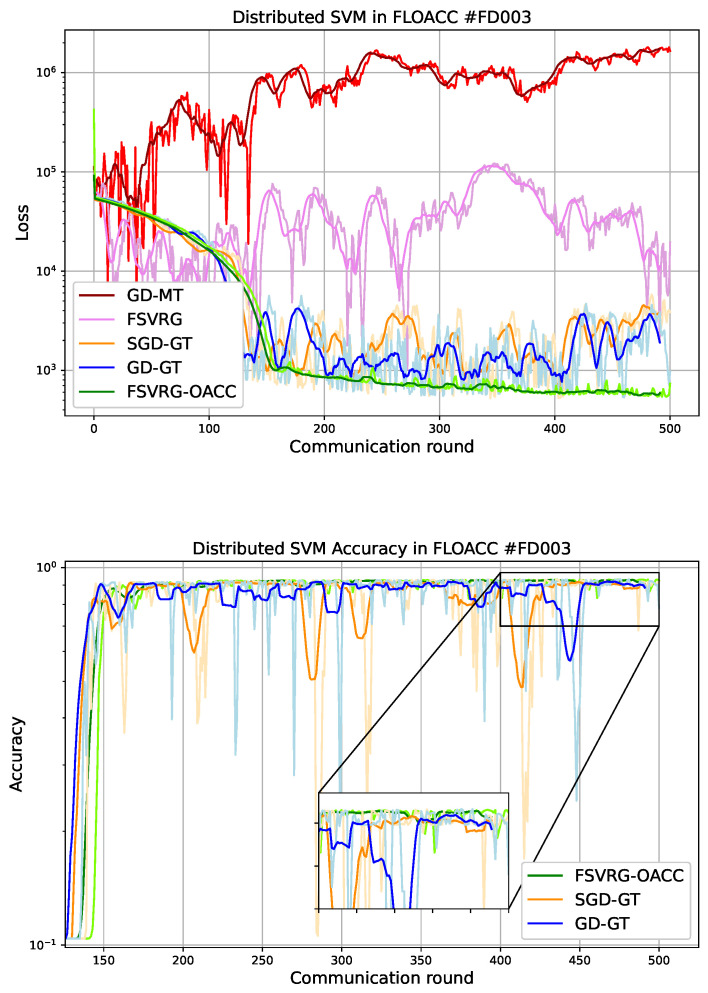
Convergence and accuracy versus communication round for different algorithms in Over-the-Air analog aggregation ($\sigma_{z}^{2}=1$, p=1, and dataset: FD003). The pale graphs represent the signal, while the bold graphs depict the windowed average of the signal.

**Figure 5 sensors-23-07840-f005:**
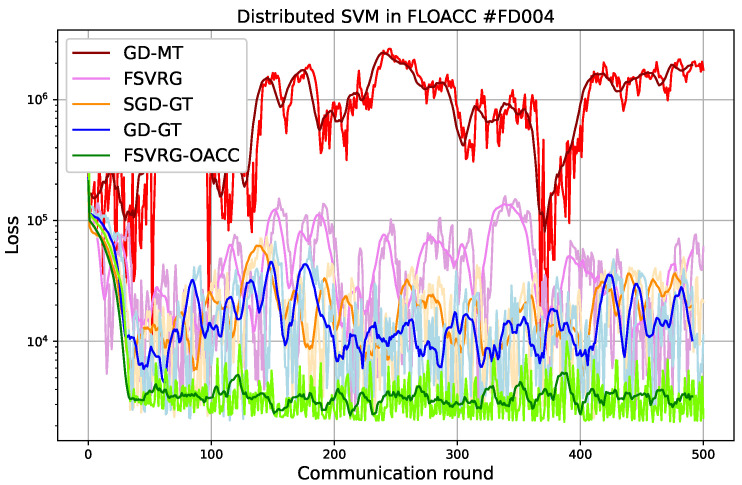
Convergence and accuracy versus communication round for different algorithms in Over-the-Air analog aggregation ($\sigma_{z}^{2}=1$, p=1, and dataset: FD004). The pale graphs represent the signal, while the bold graphs depict the windowed average of the signal.

**Figure 6 sensors-23-07840-f006:**
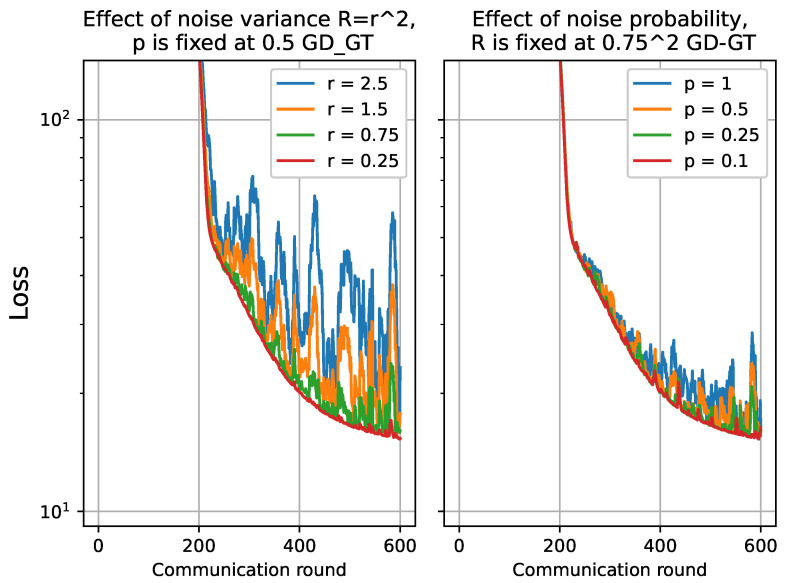
Convergence analysis for GD-GT by varying noise level $r\equiv \sigma_{z}=\left\{0.25,0.75,1.5,2.5\right\}$, and the probability of noise presence p={0.1,0.25,0.5,1}.

**Figure 7 sensors-23-07840-f007:**
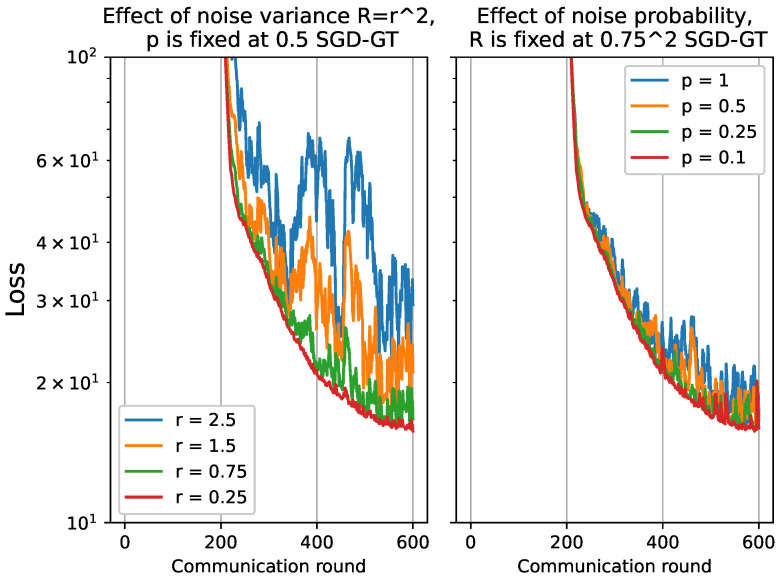
Convergence analysis for SGD-GT by varying noise level $r\equiv \sigma_{z}=\left\{0.25,0.75,1.5,2.5\right\}$, and the probability of noise presence p={0.1,0.25,0.5,1}.

**Figure 8 sensors-23-07840-f008:**
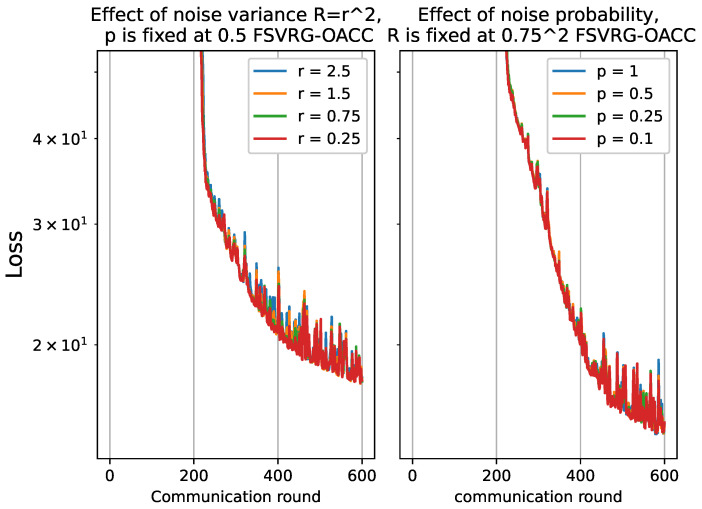
Convergence analysis for FSVRG-OACC by varying noise level $r\equiv \sigma_{z}=\left\{0.25,0.75,1.5,2.5\right\}$, and the probability of noise presence p={0.1,0.25,0.5,1}.

**Table 1 sensors-23-07840-t001:** Several examples of loss functions.

Model	Loss Function
Linear regression	${∥1-y_{j}w^{T}x_{j}∥}^{2}$
Logistic regression	$-\log\left(1+\exp\left(-y_{j}w^{T}x_{j}\right)\right)$
K-means	$\min_{l}{∥x_{j}-w_{l}∥}^{2}$
Cross-Entropy	$-\sum y_{c}p\left(y=c|x,w\right)$
Squared-SVM	${\lambda ∥w∥}^{2}+\max0;1-y_{j}w^{T}x_{j}$

**Table 2 sensors-23-07840-t002:** Performance analysis of different algorithms on FLOACC.

Dataset	Evaluation Metrics	Optimizer		
		GD-GT	SGD-GT	FSVRG-OACC
FD003	Runtime	45.8 s	21.3 s	66.7 s
	Final accuracy	76%	61%	91%
FD004	Runtime	70 s	20 s	105 s
	Final accuracy	42%	41%	61%

**Table 3 sensors-23-07840-t003:** Performance analysis of FL algorithm on noiseless communication channel [1].

Dataset	Evaluation Metrics	Optimizer		
		GD-GT	SGD-GT	FSVRG
FD003	Runtime	69 s	20.5 s	161 s
	Final accuracy	94.2%	92.2%	91.7%
FD004	Runtime	143.5 s	45.5 s	337 s
	Final accuracy	78.4%	71.7%	86.6%

## Data Availability

The CMAPSS dataset is available at https://data.nasa.gov/dataset/C-MAPSS-Aircraft-Engine-Simulator-Data/xaut-bemq (accessed on 19 August 2023).

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
