# Peer review of "Low-Latency Collaborative Predictive Maintenance: Over-the-Air Federated Learning in Noisy Industrial Environments"

_sensors, 2023, doi:10.3390/s23187840_

Round 1

Reviewer 1 Report

1. For the industrial systems, such as the Industrial Internet of Things, the authors should refer to the following recent work for giving more discussions about the prospect of future Industry 4.0 in introduction part.

--- Digital Twin-Assisted Edge Computation Offloading in Industrial Internet of Things With NOMA, IEEE Transactions on Vehicular Technology, April 2023, DOI: 10.1109/TVT.2023.3270859

2. The abstract part is too long, please reduce it.

3. Signaling and complexity of the proposed algorithm can be discussed.

4. More comparable algorithms should be considered. Some newest ones can be found in GitHub.

5. The paper should evaluate the adverse effect due to non-IID of the dataset at the local devices. Some performance evaluation should be given.

6. The proposed scheme assumes that all complete knowledge about the system and local devices is available. However, in reality, this assumption is rarely held. The paper should discuss the situation when partial information is available and also potential solutions to overcome this challenge.

None.

Author Response

All comments have been addressed in a PDF file and attached here.

Regards,/Ali

Reviewer 2 Report

Comments on the paper:
====================
1- The paper addresses a crucial issue in the emerging field of Industry 4.0 by targeting the latency in communication which is a significant bottleneck for fast edge learning in time-sensitive applications like predictive maintenance. The use of Federated Learning (FL) for preserving privacy and enabling efficient aggregation is well justified and relevant to the current industrial scenario.

2- The approach of using analog aggregation over-the-air of updates transmitted by devices simultaneously over wireless channels is innovative and has the potential to drastically reduce communication latency. However, it is essential to consider that the performance degradation due to channel noise and fading could be a significant challenge, and it is commendable that the paper addresses this issue by proposing a novel tracking-based stochastic approximation scheme integrated into a standard federated stochastic variance reduced gradient (FSVRG).

3- The numerical results and simulation experiments provided in the paper offer strong validation for the proposed approach, which is essential for building confidence in the methodology. The precise convergence guarantees without requiring an increase in the transmission power gain is a significant achievement.

4- The conclusion summarizes the key findings and contributions effectively and provides a clear direction for future work by suggesting the incorporation of power control and device selection policies into the design of the proposed algorithm.
========================
Suggestions for improvement:
========================
1- While the paper provides a comprehensive approach to address communication latency and the impact of channel noise, it would be beneficial to include a comparison with other existing methods or approaches that address similar challenges. This would provide a clearer picture of the advantages and potential limitations of the proposed approach.

2- The paper could also benefit from a more detailed discussion on the practical implications and potential challenges of implementing the proposed approach in real-world industrial environments. For example, considering the hardware and software requirements, and potential challenges related to interoperability with existing systems.

3- It would be interesting to see how the proposed approach performs in different industrial scenarios and environments, as this would provide a more comprehensive validation of the method. If possible, conducting experiments in actual industrial settings or using real-world data from different industrial sectors would provide additional validation of the approach.

4- Abstract: More than 250 words are needed to decrease

Minor editing of English language required

Author Response

(The authors gave the same response as above.)

Round 2

Reviewer 1 Report

All the concerns raised by the reviewer have been dealt with. The paper can be accepted in current form.